# Substantiation for the Use of Curcumin during the Development of Neurodegeneration after Brain Ischemia

**DOI:** 10.3390/ijms21020517

**Published:** 2020-01-14

**Authors:** Marzena Ułamek-Kozioł, Stanisław J. Czuczwar, Sławomir Januszewski, Ryszard Pluta

**Affiliations:** 1Laboratory of Ischemic and Neurodegenerative Brain Research, Mossakowski Medical Research Centre, Polish Academy of Sciences, 02-106 Warsaw, Poland; mulamek@imdik.pan.pl (M.U.-K.);; 2First Department of Neurology, Institute of Psychiatry and Neurology, 02-957 Warsaw, Poland; 3Department of Pathophysiology, Medical University of Lublin, 20-090 Lublin, Poland; czuczwarsj@yahoo.com

**Keywords:** brain ischemia, curcumin, amyloid, tau protein, neuroinflammation, apoptosis, autophagy, neurodegeneration, neuroprotection, neurogenesis

## Abstract

Currently available pharmacological treatment of post-ischemia-reperfusion brain injury has limited effectiveness. This review provides an assessment of the current state of neurodegeneration treatment due to ischemia-reperfusion brain injury and focuses on the role of curcumin in the diet. The purpose of this review was to provide a comprehensive overview of what was published about the benefits of curcumin influence on post-ischemic brain damage. Some data on the clinical benefits of curcumin treatment of post-ischemic brain in terms of clinical symptoms and adverse reactions have been reviewed. The data in this review contributes to a better understanding of the potential benefits of curcumin in the treatment of neurodegenerative changes after ischemia and informs scientists, clinicians, and patients, as well as their families and caregivers about the possibilities of such treatment. Due to the pleotropic properties of curcumin, including anti-amyloid, anti-tau protein hyperphosphorylation, anti-inflammatory, anti-apoptotic, and neuroprotective action, as well as increasing neuronal lifespan and promoting neurogenesis, curcumin is a promising candidate for the treatment of post-ischemic neurodegeneration with misfolded proteins accumulation. In this way, it may gain interest as a potential therapy to prevent the development of neurodegenerative changes after cerebral ischemia. In addition, it is a safe substance and inexpensive, easily accessible, and can effectively penetrate the blood–brain barrier and neuronal membranes. In conclusion, the evidence available in a review of the literature on the therapeutic potential of curcumin provides helpful insight into the potential clinical utility of curcumin in the treatment of neurological neurodegenerative diseases with misfolded proteins. Therefore, curcumin may be a promising supplementary agent against development of neurodegeneration after brain ischemia in the future. Indeed, there is a rational scientific basis for the use of curcumin for the prophylaxis and treatment of post-ischemic neurodegeneration.

## 1. Introduction

Ischemia-related brain injury is increasingly common in aging societies in both developed and developing countries. Ischemia-reperfusion injury of the brain in humans is the second cause of death and the third cause of disability, which may soon become the main cause of dementia [1,2,3,4]. Recent epidemiological data indicate that about 17 million patients suffer from ischemic stroke per year, of which 6 million die each year [4,5]. In the world, the number of people after cerebral ischemia has now reached about 33 million [4,5]. According to current forecasts, the number of patients will increase to 77 million in 2030 [4,5]. In 2010, the annual cost of managing ischemic stroke in Europe was around 64 billion Euros [4].

Post-stroke neurological deficits usually improve to a greater or lesser degree, while cognitive impairment gradually progresses. Prevalence of dementia after the first ischemic stroke is estimated at 10%, and after a repeated stroke at about 41% in survivors [4]. In long-term post-stroke dementia studies, the cumulative incidence over 25 years was estimated to be 48% [4]. If the trend of ischemic stroke persists, about 12 million patients will die by 2030, 70 million will be after the stroke, and over 200 million disability-adjusted life-years loss will be recorded worldwide each year [4]. Thrombolysis is currently the use of choice as a treatment during ischemic stroke in humans, but thrombolysis has a limited therapeutic window and does not affect the progressive changes that develop slowly during recirculation [6]. Brain ischemia patients, as well as experimental animals develop cognitive deficits depending on survival [1,2,3,4,7,8,9,10]. Currently, the important role of episodic brain ischemia in the induction of dementia is a priority in both experimental and clinical research [3,4,11]. New research suggests that brain damage as a result of ischemia-reperfusion causes neurodegeneration of the brain through the development of inflammation [12,13,14,15], the generation and accumulation of various parts of the amyloid protein precursor [4,14,16] and tau protein dysfunction [17,18], which in turn damage neurons, especially in different regions of the hippocampus and ultimately contribute to brain atrophy [11,19,20,21,22]. Additionally, local brain ischemia in humans increases the production and accumulation of amyloid in the brain, as confirmed by positron emission tomography [4]. On the other hand, experimental studies have revealed that high levels of amyloid in the brain after ischemia increase the extent of the infarction [4]. Several years of intensive research have revealed that ischemic stroke and experimental cerebral ischemia are associated with numerous neuronal changes, including mitochondrial damage, synapse disappearance, β-amyloid peptide production and accumulation, microglia and astrocyte activation, tau protein phosphorylation, and neurofibrillary tangles formation [4,14,15,17,18,19,23,24]. However, we do not have drugs/agents that prevent brain ischemia and/or can delay or stop the progression of neurodegeneration after ischemia.

In the absence of translation of experimental neuroprotective molecules used in animals for use in clinical settings [25], we focus our care on improving motor and cognitive functions post-ischemia, and not on protecting neuronal cells during ischemia. That is why we are forced to improve the activity of persistent neurons and cognitive function after ischemic episode [3,7,8,9,10] and new treatment should be designed in this way to improve functional activities after cerebral ischemia by effectively extending the therapeutic window post-ischemia. It should be emphasized that, despite the fact that ischemic stroke is one of the main causes of death and disability in the world, there is currently no effective treatment that would improve structural and functional changes ultimately leading to neurodegeneration with subsequent dementia. Therefore, in this review, we will focus on the protective effect of pleiotropic curcumin on persistent neurons and pathological consequences that develop after an ischemic stroke.

## 2. Neuropathology after Brain Ischemia

In the hippocampus and third, fifth, and sixth layer of the cortex, necrotic and apoptotic neuronal cells were superimposed with injured neurons during seven days after ischemia-reperfusion brain injury [11,14,19]. In later recirculation times up to six months, the number of injured neuronal cells was reduced and loss of neurons predominated. After survival longer than six months following ischemia-reperfusion brain injury, in addition to local neuronal loss, acute and chronic neuronal changes were noted. Changes occurred in regions of the brain that were not involved in early changes, such as the CA2, CA3, and CA4 areas of the hippocampus [11,14,19].

The loss of neuronal cells in the hippocampus with a decrease in acetylcholine levels was evident after cerebral ischemia, suggesting that the loss of neuronal cells may be due to insufficient neuronal excitable transmission [26,27]. Reduced levels of both synaptophysin and 95-density postsynaptic protein were observed in the hippocampus simultaneously with ultrastructural synaptic changes after focal cerebral ischemia [28,29]. Other investigations have presented that ischemia-reperfusion brain injury leads to stimulation of synaptic autophagy, which may be associated with neuronal death in the hippocampus following transient brain ischemia [26,30,31,32,33]. Intracellular Ca^2+^ increase after ischemia [26] regulates calpain activity in neuronal cells whose target proteins are found in GABAergic and glutaminergic synapses. After cerebral ischemia, calpain cleaves pre- and postsynaptic proteins, and calpain-cleaved proteins contribute to the death of ischemic neurons [34].

Alterations in white matter and activation of glial cells have been reported in both humans and animals following ischemia-reperfusion brain injury [14,15,19,20,21,35,36,37]. In experimental models of transient cerebral ischemia, ischemia causes serious changes in both the corpus callosum and subcortical white matter [14,20,21,38]. These changes are coherent with the activation of glial cells in the corpus callosum after brain damage due to ischemia-reperfusion [39]. Brain ischemia increases the permeability of the blood–brain barrier, which facilitates the invasion of inflammatory cells into the brain tissue and leakage of amyloid and tau protein from serum into the brain parenchyma, which in turn leads to progressive damage to white matter [15,20,21,40,41,42,43,44,45,46,47,48,49,50,51,52,53].

Ischemic alterations in the brain showed symptoms of a gradually progressive neurodegenerative process that develops slowly over a long period of time during recirculation following an episode of cerebral ischemia [14]. An autopsy study of the brain carried out within 1–2 years after ischemic injury, presented sings of hydrocephalus of the brain [14,19,22] with widening of the subarachnoid space around the cerebral hemispheres [19]. During this time, total hippocampus atrophy with very narrow cerebral cortex dominated [14,19,22]. The late features of atrophy of the white matter of the brain were manifested as scattered changes in the form of cavitations, and the rarefaction was manifested as an advanced spongiosis. This phenomenon can be explained by mass neuronal death along with chronic pathological post-ischemic increased blood-brain barrier permeability [20,31,44,47] for amyloid and tau protein [41,42,43,46,48,49,50,51,52,53].

## 3. Amyloid after Brain Ischemia

Brains up to six months after ischemia revealed staining in the intra- and extracellular space at the N- and C-terminal of the amyloid protein precursor, as well as β-amyloid peptide [16,19,22,54,55,56,57,58,59,60,61,62,63,64,65,66,67]. Experimental ischemic brain injury, with a survival of 6–12 months, showed staining in the intra- and extracellular space of the brain parenchyma only at the C-terminal of the amyloid protein precursor and β-amyloid peptide [14,19,22,60]. Staining fragments of the amyloid protein precursor were found in neurons, microglial cells, astrocytes, and oligodendrocytes [19,58,63,68,69,70,71,72]. Reactive astrocytes with abnormal accumulation of β-amyloid peptide may be involved in the dangerous repair of ischemic tissue accompanied by death of astrocytes [16,19,40,63,73,74]. Astrocytes with a huge accumulation of various parts of the amyloid protein precursor may be involved in the development of glial scar after cerebral ischemia [19,70,71,72,73]. Staining at the N- and C-terminal of the amyloid protein precursor and β-amyloid peptide was found in periventricular and subcortical white matter after an episode of cerebral ischemia [14,20,21]. It was noted that the more intense the damage to white matter after ischemia, the more intense the staining of all fragments of the amyloid protein precursor in the brain [75]. It is suggested that the above changes are responsible for the development of leukoaraiosis after ischemic brain damage [21]. The extracellular accumulations of different fragments of the amyloid protein precursor had the character of very small dots up to the characteristic diffuse and dense amyloid plaque-like deposits [12,13,14,16,19,43,44,47,48,60,63,76,77]. Diffuse amyloid plaques were observed in the ischemic cortex, entorhinal cortex, hippocampus, corpus callosum, and around the lateral ventricles but dense amyloid plaque-like deposits were noted only in post-ischemic thalamus [76]. Deposition of various parts of the amyloid protein precursor in neurons, especially β-amyloid peptide, and in astrocytes, indicates the important role of the amyloid protein precursor in the development of neurodegenerative processes after cerebral ischemia [12,13,16,57,71,72]. These deposits affecting neurons in various brain regions cause synaptic degradation and activate additional retrograde neuronal cell death after ischemia [78]. Data indicate that the post-ischemic C-terminal of amyloid protein precursor and β-amyloid peptide may be responsible for secondary neurodegenerative processes that worsen post-ischemic outcome due to progressive neuronal death [8,14,22,59,60,66,67,79,80,81]. Data also demonstrate that after experimental cerebral ischemia, β-amyloid peptide is generated as a result of neuronal injury and death [56], which further contributes to the development of dementia with the Alzheimer’s disease phenotype through neurotoxic effects [67,82,83].

In humans after brain ischemia-reperfusion damage, amyloid deposits in different structures of the brain have been observed [84,85,86,87]. Studies have shown both diffuse and senile amyloid plaques in regions of the brain prone to ischemia, in the brain cortex, and in arterial border zones [84,85,86,87]. In the perivascular space of grey and white matter amyloid was noted, too [85]. Neurons from the hippocampus and cortex were strongly stained on amyloid. Moreover, epithelial and ependymal cells were stained on amyloid. According to Qi et al. [24] the staining of amyloid 1–40 and 1–42 in the human ischemic hippocampus was observed in the intra- and extracellular space. After brain ischemia cells of the epithelium, choroid plexus and the cells bordering the brain ventricles were stained for receptor for advanced glycation end products [87]. In addition, clinical study presented an increase of amyloid in blood in patients following ischemic brain injury [46,50].

## 4. Tau Protein after Brain Ischemia

Intensive staining of tau protein in neurons, astrocytes, microglial cells, and oligodendrocytes after ischemia was observed in both the hippocampus and cortex [64,88,89,90,91,92,93,94,95,96]. Another study showed that the tau protein can inhibit the transport of the amyloid protein precursor in axons and dendrites, which leads to the deposition of the amyloid protein precursor in the body of a neuronal cell [97]. Available studies demonstrate that following ischemia, hyperphosphorylated tau protein predominates in cortical neurons and accompanies apoptosis [17,95,96,98,99]. The above data indicate that neuronal apoptosis after brain damage due to ischemia-reperfusion is directly associated with hyperphosphorylated tau protein. Hyperphosphorylated tau protein deposited as paired helical filaments in brain tissue and associated with memory dysfunction after global brain ischemia in mice [100]. Wen et al. [17,98,99] revealed that reversible brain damage due to ischemia-reperfusion was involved in the development of neurofibrillary tangle-like after experimental local brain ischemia. In addition, Kato et al. [23] presented in people the formation of neurofibrillary tangle in the Meynert nucleus ipsilateral in relation to the area of cerebral infarction. The level of tau protein was detectable in plasma after complete brain ischemia in humans within 96 h and probably indicated the progression of neuronal changes during recirculation [49,51]. The presented investigations suggest that the level of tau protein in the blood can be used as an indicator of the neurological outcome after brain damage by ischemia-reperfusion [49,51].

## 5. Reasons for Using Curcumin after Brain Ischemia

Ischemic stroke in humans and animals is mainly age-related and is characterized by various neurobehavioral changes, but primarily memory deficits, with a gradual decline in cognitive and intellectual functions that ultimately lead to dementia. In humans, this is the main cause of death for the elderly. New features of ischemic stroke in humans and experimental cerebral ischemia are the accumulation of amyloid as diffuse and senile plaques in the extracellular space [14,16,24,76,85,87] and hyperphosphorylated tau protein [100] as neurofibrillary tangle in neurons [23]. The accumulation of these incorrectly folded proteins is thought to be additional cause of neuronal death, synaptic loss, oxidative damage, and the development of neuroinflammation [15] in the brain after cerebral ischemia. Therefore, substances with pleiotropic activity, especially those with anti-amyloid and anti-tau protein properties, and those that reduce oxidative damage and inflammation should provide the greatest potential for preventing ischemic stroke and/or loss of neurons observed in the brain after ischemia. Recently, curcumin is considered one of the interesting and most promising natural pleiotropic molecules for use in the treatment of ischemic stroke due to its pleiotropic effects.

Curcumin is a neuroprotective molecule with strong antioxidant and anti-inflammatory properties [101,102]. Its anti-amyloid properties make it the most promising compound for the treatment of various brain diseases with the accumulation of folding proteins. In addition, curcumin is both hydrophobic and lipophilic, and because the brain contains large amounts of lipids, the absorption, bioavailability and half-life profiles of curcumin are very beneficial in the brain. Several evidences have shown that inflammation, oxidative damage, and the accumulation of misfolded amyloid proteins synergistically contribute to brain damage due to ischemia and reperfusion [4,11,18]. Therefore, targeting these pathways may be the basic strategy for developing post-ischemic neurodegeneration therapy. In this context, the use of curcumin in the treatment of post-ischemic neurodegeneration has certain advantages, namely (1) it easily crosses the blood-brain barrier [103,104], (2) binds and disaggregates oligomers and fibrils of amyloid proteins [105,106], (3) increases the clearance of amyloid [107], (4) reduces chronic inflammation, (5) acts as a powerful antioxidant, (6) stimulates neurogenesis, (7) removes metal from amyloid, (8) can be taken in relatively high doses without side effects [108].

### 5.1. Neuroprotection and Neurogenesis

In animal focal cerebral ischemia, curcumin reduced both the volume of brain infarction [109,110,111,112,113,114,115,116,117,118,119] and brain edema [109,110,113,114,115,120,121] and gave better neurological outcomes than in the vehicle group [112,114,117,122]. Curcumin improved sensory, motor, and locomotor activity, both smaller neurobehavioral and neurological deficits [111,115,117,119,120,123,124]. Curcumin also reduced neuronal apoptosis by increasing the anti-apoptotic Bcl2 protein at the mitochondrial level, lowered cytosolic cytochrome c translocation [112,118] and decreased the mitochondrial membrane potential [121,125]. In addition, it causes reduced apoptosis [124,125,126] via caspase-3 mRNA downregulation [117,119] and stimulates neurogenesis [123] (Figure 1). Curcumin also reduces post-ischemic circulatory disturbances induced by neutrophils in animal stroke by preventing neutrophil adhesion to the brain microcirculation [114]. Moreover, it reduces blood–brain barrier permeability [110,113], glial activation [124], neuroinflammation and astrogliosis [115]. In the same model, curcumin administration reduced neuroinflammation by lowering the IL6, TNFα [117,121], and iNOS levels, and mediating autophagy activity via PI3K/Akt/mTOR [117,119]. Results similar to those described for rats were also obtained in a mouse model of local cerebral ischemia, in which administration of curcumin reduced cerebral infarction and neuronal apoptosis most likely by limitation of mitochondrial dysfunction [127]. In addition, curcumin has been found to improve neurological function score, maintain the integrity of the blood–brain barrier, and reduce the infarct volume of the cerebral cortex, and decrease mortality, as well as apoptosis of neurons in animal brain ischemia models [128,129,130]. Further, curcumin attenuates glutamate neurotoxicity in the hippocampus [128]. In a recent study, curcumin protected ischemic neurons from death and apoptosis through the neuroprotective effect of curcumin associated with both HIF-1α and autophagy inhibition [131].

### 5.2. Inhibition of Amyloid Production

The β-amyloid peptide is a product of transmembrane protein metabolism, called the amyloid protein precursor. The generation of β-amyloid peptide is catalyzed by two enzymes, first by β-secretase and then by γ-secretase. It is suggested that during post-ischemia progression, the induction of neuroinflammatory signals increases β-amyloid peptide production by increasing β-secretase activity [132], while curcumin inhibits β-secretase activity, thereby lowering β-amyloid peptide levels [102,105,132] (Figure 1). In addition, curcumin is a potent inhibitor of the maturation of the amyloid protein precursor and the amyloidogenic pathway, which contributes to the reduction of β-amyloid peptide levels [133,134] (Figure 1). In addition, curcumin may regulate β-amyloid peptide generation by inhibiting GSK-3β-mediated presenilin-1 activity, which is one of the components of γ-secretase [135].

It has been presented that amyloid production can be limited by mechanisms such as metal chelation [136] and reduction of β-secretase induction by pro-inflammatory mediators [102]. β-secretase is thought to play a major role in initiating amyloid production [137], and is therefore an attractive target for curcumin after cerebral ischemia. Zhang et al. [133] suggested modulating levels of amyloid protein precursor in the secretory pathway as a cellular mechanism by which curcumin reduces amyloid levels (Figure 1). In addition, they noted that the use of curcumin significantly increased the retention of the immature amyloid protein precursor in the endoplasmic reticulum. On top of it, the authors suggested that curcumin may interfere with endocytosis of the amyloid protein precursor [133].

### 5.3. Inhibition of Amyloid Aggregation

Experimental evidences have shown that curcumin can directly bind to the β-pleated sheet structures of amyloid [138]. Interestingly, curcumin exhibits the most potent inhibitory effect on amyloid aggregation (Figure 1) amongst 214 antioxidant substances evaluated in vitro [139,140,141,142], indicating that it is one of the strongest anti-amyloid compounds tested so far. In vitro and in vivo studies have shown that curcumin has a dose-dependent effect on the inhibition of amyloid peptide 1–40 and 1–42 fibrils formation, with an EC50 of 0.09–0.63 µM [105,143]. Several investigations have shown that curcumin can reduce the accumulation of oligomers of both amyloid peptide 1–40 and 1–42, as well as the formation of fibrils [105,143,144]. Curcumin attenuates β-amyloid peptide 1–40 aggregate toxicity and modulates its aggregation pathway [145] (Figure 1). The above study showed that curcumin did not inhibit the formation of amyloid fibrils, but rather enriched the population of soluble “off-pathway” oligomers and pre-fibrillary aggregates that are not toxic. Curcumin also exerted a nonspecific neuroprotective effect, reducing the toxicity induced by many amyloid conformers, including monomeric, oligomeric, prefibrillar, and fibrillar amyloid. The curcumin-mediated neuroprotective effect occurs because curcumin reduced the permeability of the cell membrane induced by amyloid aggregates. In conclusion, this study showed that curcumin exerted a neuroprotective effect against amyloid-induced toxicity through at least two compatible pathways, modifying the amyloid aggregation pathway towards the formation of nontoxic aggregates (Figure 1) and alleviating amyloid-induced toxicity, probably via a nonspecific pathway [145].

Brahmkhatri et al. [146] observed that curcumin-laden gold nanoparticles inhibited aggregation of N-terminal area of an amyloid and were able to dissolve aggregates. Similarly, Mithu et al. [147] found that curcumin disorganized amyloid fibrils—the disruption of amyloid fibers occurred by means of structural changes in the salt bridge area and near the C-terminal of amyloid protein precursor. A more detailed study on inhibition of amyloid aggregation has shown that in addition to curcumin that inhibits fibril formation in vitro, it also inhibits the formation of amyloid oligomers, fibrils, binds plaques, disrupts existing plaques, and reduces amyloid level and toxicity in vivo [105,145,148] (Figure 1). In addition, systemic administration of curcumin to transgenic mice for seven days cleared and reduced existing amyloid plaques, which was monitored by longitudinal imaging, which indicated a strong disaggregation effect. Curcumin also caused a limited but significant reversal of structural changes in dystrophic dendrites. Together, these results suggest that curcumin reverses existing amyloid pathology and related neurotoxicity in transgenic mice [148].

### 5.4. β-Amyloid Peptide Clearance

The level of β-amyloid peptide in the brain of ischemic stroke in patients depends on the balance between generation, clearance, and influx of β-amyloid peptide from the blood. Disruption of clearance ways promotes the growth of β-amyloid peptide in brain tissue. However, there are several pathways to remove β-amyloid peptide from a neuron, including transport of β-amyloid peptide via LRP1 across the blood–brain barrier to the blood, followed by enzymes that degrade β-amyloid peptide, as well as by involving the immune system [149]. Curcumin acts in a similar way to the amyloid vaccine [102] and may bind to β-amyloid peptide to allow its removal from brain tissue by promoting receptor-mediated efflux of β-amyloid peptide [105]. On the other hand, curcumin can reduce the brain tissue load of β-amyloid peptide by suppressing RAGE-mediated influx of β-amyloid peptide across the blood–brain barrier from blood and by enhancing the enzymatic degradation of β-amyloid peptide. In addition, curcumin may stimulate phagocytosis [108] and increase the presence of phagocytic cells around deposits of β-amyloid peptide, as observed in various experimental models [150], as well as with amyloid plaques in sections of post-mortem human brain exposed to primary rodent microglia [151]. Curcumin stimulates amyloid phagocytosis by activating microglial cells and enzymatic breakdown of amyloid [108]. Curcumin also stimulates B lymphocytes to trigger an amyloid-specific antibody that puts out of action amyloid. In summary, curcumin simultaneously inhibits the influx of amyloid from the blood into the brain tissue and increases its efflux from the brain into the circulation (Figure 1).

### 5.5. Inhibition of Tau Protein Phosphorylation

The second most common pathology associated with protein folding and observed in ischemic stroke and experimental brain ischemia are neurofibrillary tangle, which are essentially associated with the deposition of hyperphosphorylated tau protein as paired helical filaments [17,23,98,99,100]. The tau protein is a microtubule stabilizing protein and neurons are rich in this protein. Tau protein hyperphosphorylation causes cytoarchitectonic changes that develop oxidative stress, mitochondrial dysfunction, and progression of neurodegeneration [152]. Tau protein hyperphosphorylation and deposition as neurofibrillary tangle are regulated by several tau protein kinases, the most common of which is glycogen synthase kinase-3β and mitogen activated protein kinase [153]. The common tau protein kinases, which can phosphorylate the tau protein are cyclin-dependent kinase 5, extracellular signal-regulated kinase 2, S6 kinase, microtubule affinity-regulating kinase, SAD kinase, protein kinase A, calcium/calmodulin-dependent protein kinase II, or Src family kinases, such as Fyn and c-Abl. Thus, inhibition of tau protein kinases appears to be a viable strategy for preventing neurodegeneration induced by neurofibrillary tangle. Curcumin has been found to bind to neurofibrillary tangle in the brain of Alzheimer’s disease, as well as in the brains of experimental Alzheimer’s disease, which inhibits the accumulation of prion protein [154]. An in vitro experiment presented that curcumin inhibits hyperphosphorylated tau protein aggregation and disintegrates performed tau protein filaments [155]. Curcumin has been shown to inhibit glycogen synthase kinase-3β activity and reduce tau protein dimmer formation and hyperphosphorylated tau protein oligomerization in aged tau protein transgenic mice [156]. On the other hand, oral administration of curcumin with docosahexaenoic acid reduced hyperphosphorylated tau protein levels by inhibiting insulin receptor substrate 1 and c-Jun N-terminal kinase activities [156] (Figure 1).

## 6. Limitations of Curcumin Treatment and Side Effects

Based on previous clinical studies, 8 g of short-term curcumin daily intake has been shown to have no significant side effects [157]. Another phase 1 human study, with 8 g of curcumin daily for three months, did not reveal any toxic effects [144]. Toxicological assessments have shown that curcumin is considered a pharmacologically safe substance, up to 12 g daily, as shown in animal studies and phase 1 clinical study [103,157]. On the other hand, doses of curcumin above 8 g daily were unacceptable to patients due to the large volume of tablets [157]. Extrapolation of animal investigations to clinical studies has shown that in order to obtain beneficial effects in humans, oral curcumin intake in the range of 80–500 mg daily is recommended, which means that the daily intake of raw curcumin would be 2–4 g [158,159]. However, for therapeutic purposes, curcumin is very unstable in most body fluids, and due to its poor water solubility, it is recommended to mix curcumin with milk or oil to increase its absorption and metabolism [108].

However, several studies have shown that high doses of curcumin can cause adverse side effects including headache, nausea, gastrointestinal discomfort, intractable abdominal pain, yellow stool, skin rash, swelling of the skin, diarrhea, chest tightness, as well as allergic reactions such as dermatitis [157,160]. In human curcumin studies, doses of 0.9–3.6 g daily for 1–4 months caused adverse reactions, including diarrhea and nausea, with increases in serum lactate dehydrogenase and alkaline phosphatase [161]. Chronic curcumin intake can sometimes be hepatotoxic. Therefore, people with liver diseases such as cirrhosis, gallstones, gallbladder and biliary tract obstruction, acute biliary colic and obstructive jaundice, or those who use drugs for liver problems are contraindicated in curcumin treatment because curcumin may stimulate bile secretion [162]. In fact, a dose of up to 20–40 mg of curcumin a day can increase gallbladder contractions in healthy people [163,164]. Similarly, alcoholics or heavy drinkers cannot use curcumin. In addition, curcumin is not recommended for people taking blood thinners, reserpine, or nonsteroid anti-inflammatory drugs, as it may interact with these drugs [102,158,159]. Therefore, more research is needed to assess the long-term toxicity/side effects associated with curcumin before it can be approved for clinical use.

## 7. Conclusions

Neuronal damage and death, with misfolded protein deposits, as well as cognitive deficits and/or impairment of motor coordination with development of dementia of Alzheimer’s disease phenotype are the main problems in brain damage due to ischemia-reperfusion [165]. Due to the pleotropic influence of curcumin on the brain, including anti-amyloid, anti-tau protein, anti-inflammatory, and neuroprotective properties, curcumin is a promising candidate for the treatment of post-ischemic neurodegeneration with misfolded protein deposits. In addition, it is a safe substance and inexpensive, easily accessible and can effectively penetrate the blood–brain barrier and neuronal membranes. In conclusion, the evidence available in a review of the literature on the therapeutic potential of curcumin provide helpful insight into the potential clinical utility of curcumin in the treatment of neurological diseases with misfolded proteins.

## 8. Outlook

The data shown in this review indicate promising results in the use of curcumin in post-ischemic neurodegeneration. However, a limited number of studies provide evidence of overall quality from low to very low. Some studies have shown side effects, although the study times were short, so the long-term risk associated with these side effects is unknown. Future randomized clinical trials are needed to confirm the efficacy of curcumin and provide further information on some unresolved practical problems, i.e., how long curcumin can be given. What is more, these questions can only be answered by undertaking large, multi-center research efforts.

To date, the data of using curcumin as a drug in the therapy of post-ischemic brain injury seem particularly interesting due to the effect on the accumulation of amyloid and tau protein, despite a very limited number of studies. The few human investigations available to date are without an authentic reference control and randomization group. The results shown causal evidence and underline the need for further research to show that curcumin beneficially affects the brain after ischemia. In recent years, curcumin’s reputation for therapeutic effects has grown significantly. Curcumin gives misleading results in molecular and partly polemic publications, because this substance has no single target. In addition, the fact that curcumin, like many other natural substances, has more than one drug target, indicates its versatile use and low risk of inducing resistance to treatment. Although there are reasonable doubts which are important for the reliability of therapeutic results, it makes no sense to disparage everything that has been published so far about the therapeutic effects of curcumin in the therapy of neurodegeneration after brain ischemia. Instead, we should take the challenge of distinguishing between scientifically valid and doubtful data; otherwise, we will lose the promising substance necessary for complementary and alternative therapy strategies in this case during post-ischemic neurodegeneration. Rejecting some of the effects of curcumin treatment would simply be ignorance. Opponents of curcumin treatment criticize the fact that its effectiveness has never been demonstrated in a randomized, double-blind, placebo-controlled clinical trial. Critics should consider the fact that it is difficult to obtain financial support for conducting a clinical trial with a substance that cannot be patented and that will not bring economic benefits in the future. Another issue is the design of the study, curcumin cannot be studied in randomized placebo-controlled studies, because comparing the test substance with standard therapy is currently required in clinical studies, otherwise the study will not get approval from the ethics committee. Therefore, the main question is against which drug now used to treat brain damage caused by ischemia-reperfusion should we test curcumin? There is no doubt that due to comprehensive preclinical results, as well as the first data of individual patients, the next task must be to test curcumin in well-designed and controlled clinical trials. However, the biggest challenge will be finding sponsors to financially support clinical curcumin research, and this is because this promising substance will not bring future economic benefits to potential sponsors. Still, more double-blind studies are needed to clarify the treatment options for curcumin. In summary, first, future research should focus on the proper selection of patients. In addition, a thorough and definitive explanation of the curative properties of curcumin can give hope for a long-term therapeutic effect.

Based on the results of clinical studies reviewed, it may appear that the clinical efficacy of curcumin is promisingly too good to be true. However, curcumin has not yet been approved for use in the clinic. Low bioavailability and reported adverse effects in some studies represent the main limitation of curcumin’s therapeutic utility in the clinic. We hope that the results of ongoing clinical trials help us better understand the therapeutic potential of curcumin and help put this fascinating substance at the forefront of new therapies.

## Figures and Tables

**Figure 1 ijms-21-00517-f001:**
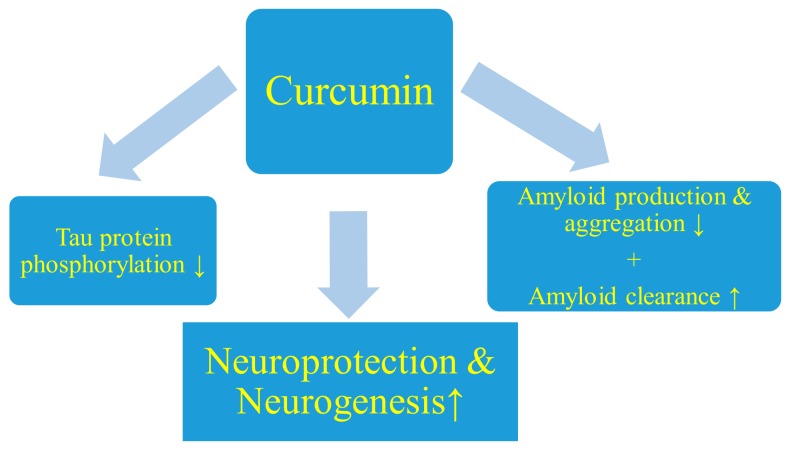
The likely effect of curcumin on post-ischemic neurodegeneration. ↓: Decrease.↑: Increase.

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
