# Peer review of "Substantiation for the Use of Curcumin during the Development of Neurodegeneration after Brain Ischemia"

_ijms, 2020, doi:10.3390/ijms21020517_

Round 1

Reviewer 1 Report

The Authors should include also a paragraph about the effects of curcumin on gene expression mechanisms.

Reviewer 2 Report

The paper by R. Pluta and coworkers is a timing review of the prospective use o curcumin for the treatment of neurodegenerative effects induced by ischemia-reperfusion in the brain. This comprehensive report will be useful to all scientists interested in neurodegenerative diseases and should be published in IJMS. However, before being accepted for publication, the manuscript needs improvement in the presentation because in the present form, without any illustration or graphical material, it is difficult to read, especially for non-expert in the field. First of all, the chemical structure of curcumin must be given, possibly mentioning that more than a single species is normally included within the name of the compound. Other schematic representations of e.g. the brain areas involved in the main diseases would help non-expert readers and make the understanding of the text easier.
